# The Linear-Logistic Model: A Novel Paradigm for Estimating Dietary Amino Acid Requirements

**DOI:** 10.3390/ani13101708

**Published:** 2023-05-22

**Authors:** Christian D. Ramirez-Camba, Crystal L. Levesque

**Affiliations:** 1Department of Animal Science, University of Minnesota, St. Paul, MN 55108, USA; ramir643@umn.edu; 2Department of Animal Science, South Dakota State University, Brookings, SD 57007, USA

**Keywords:** amino acid requirements, exploratory data analysis, functional amino acids, health, survival

## Abstract

**Simple Summary:**

This study discusses the importance of providing animals with adequate nutrients to promote their health and welfare. Current methods for estimating amino acid (AA) requirements only consider growth and protein retention, without considering other physiological responses. To address this, the authors used exploratory data analysis (EDA) to investigate whether growth and protein retention measurements are enough to develop dietary AA recommendations that optimize animal performance, health, and survival. The EDA revealed that AA dietary levels above those for maximum growth can lead to improvements in physiological responses related to milk yield, litter size, immune response, intestinal permeability, and plasma AA concentrations.

**Abstract:**

This study aimed to determine whether current methods for estimating AA requirements for animal health and welfare are sufficient. An exploratory data analysis (EDA) was conducted, which involved a review of assumptions underlying AA requirements research, a data mining approach to identify animal responses to dietary AA levels exceeding those for maximum protein retention, and a literature review to assess the physiological relevance of the linear-logistic model developed through the data mining approach. The results showed that AA dietary levels above those for maximum growth resulted in improvements in key physiological responses, and the linear-logistic model depicted the AA level at which growth and protein retention rates were maximized, along with key metabolic functions related to milk yield, litter size, immune response, intestinal permeability, and plasma AA concentrations. The results suggest that current methods based solely on growth and protein retention measurements are insufficient for optimizing key physiological responses associated with health, survival, and reproduction. The linear-logistic model could be used to estimate AA doses that optimize these responses and, potentially, survival rates.

## 1. Introduction

High mortality rates in livestock species such as swine and poultry are a major economic and welfare concern. In major pig-producing countries, the reports show that over 10% of sows die naturally or are euthanized [1,2,3,4], and nearly 30% of pigs do not make it to the market [5,6,7,8]. In poultry, especially broilers, mortality rates can range from 2% to 8% [9,10,11]. While several factors contribute to these mortality and removal rates, adequate nutrition is critical because nutrient deficiencies have been shown to worsen the problem [12]. However, what constitutes adequate nutrition in the context of survival is not clearly defined. Current methods for estimating amino acid (AA) requirements are based on growth or protein retention measurements, and there is evidence that AA levels above those required for protein retention improve key physiological functions related to animal survival [13,14,15]. This leads to the question of whether growth and protein retention measurements alone are sufficient to develop dietary AA recommendations that optimize animal performance and survival.

To answer this question, an exploratory data analysis (EDA) was performed. The EDA approach was developed by Tukey [16] and is used to conduct preliminary investigations on data to discover patterns, identify anomalies, and check assumptions using various data analytics and visualization tools. The main goal of EDA is to analyze what data can reveal beyond formal modeling or hypothesis testing. According to Tukey [16], the principles and procedures that can be called confirmatory analysis (e.g., hypothesis testing) are one of the great intellectual products of humankind, and while scientific progress is dependent on confirmatory analysis, we do not need to start with it. Accordingly, the current study focuses on EDA, leaving confirmatory analysis for future research. The first step, in accordance with EDA principles, is to examine the current assumptions and methods used for estimating amino acid requirements.

The consideration that measures of growth and/or protein retention are adequate for AA requirement estimation is based on the concept of the hierarchy of nutrient use proposed by Hammond [17]. The Hammond model describes the use of nutrients among tissues based on their priority of need, with nutrients being used first for functions and tissues essential for the individual’s survival (maintenance), then for those required for the production and survival of the animal’s offspring (reproduction), animal growth (bone and skeletal muscle deposition), and finally adipose tissue deposition (Figure 1A). Per this conceptual model, the minimal levels of dietary AA required to achieve maximum growth or protein retention are adequate to fulfill an animal’s physiological needs, with the exception of adipose tissue deposition.

Consequently, the current methods for estimating requirements are primarily concerned with determining the minimum dietary AA levels that maximize growth or protein retention. The methods used for this purpose include the indicator amino acid oxidation (IAAO) technique, nitrogen balance, and measures of body weight gain [18]. The IAAO technique is based on the concept that, when the essential AA in study is deficient for protein synthesis, the other essential AA, including the indicator (e.g., 1-^13^C-phenylalanine) will be oxidized. As a result, the IAAO technique measures the oxidation of the indicator to indirectly study patterns of body protein synthesis. Under the nitrogen balance method, the difference between nitrogen intake and nitrogen excretion reflects total body protein retention. In both methods, there is a presumed plateau in response at AA intakes above the determined breakpoint.

Other methods, such as the plasma AA technique, are also used for AA requirement estimation, as they similarly coincide with levels that maximize growth and protein retention (Figure 1B) [18]. Using the plasma AA technique, the requirement is defined as the point where an additional dietary AA supply causes an increase in plasma AA levels, based on the notion that the body lacks a storage compartment for free AA [19]. Under the plasma AA technique, increases in plasma AA levels have been regarded as waste and potentially harmful based on the observation that excess amounts of AA accumulate in the body in rare disorders such as phenylketonuria [19]. Nevertheless, there is evidence that AA intake at levels higher than those required for maximum growth (or maximum protein retention) improves various physiological functions such as protein turnover [20] and transmethylation reactions [14]. This suggests that AA levels above those required for optimal growth are not necessarily detrimental or excessive to the animal but can actually improve its metabolic status. Nevertheless, there must exist a threshold above which the consumption of AA is detrimental to the animal. Consequently, the next step of the EDA was to identify the levels of dietary AA that optimize animal metabolic status and, potentially, survival.

## 2. Materials and Methods

To identify patterns in animal response to AA intake above the threshold for maximal growth, data from AA dose titration studies in various species and age groups were mined. This data mining process was not carried out systematically because it only looked for evident patterns without having previously established criteria. The data mining approach involved primarily data visualization, which revealed a distinct and consistent pattern that led to the development of the linear-logistic model. The studies from which data were extracted and mined and which show the linear-logistic trend in multiple species are reported in the results section.

The developed model was termed linear-logistic because it describes a trend that can be mathematically described by a linear function (y=a+bx), from which a logistic function ($y=\frac{c}{1+de^{-fx}}$) is subtracted (Equation (1)). In Equation (1), *x* represents AA or protein intake, and *y* represents nitrogen or protein retention, and related responses such as body weight gain and average daily gain.
(1)$$\mathrm{Linear-logistic}\;\mathrm{model}:\;y=a+bx-\frac{c}{{1+de}^{-fx}}$$

The linear-logistic model depicts two inflection points, the inflection point where the response is maximized and the inflection point where the response is minimized (Rmax and Rmin, respectively; Figure 2).

It is mathematically possible to calculate Rmax and Rmin by setting the first derivative (i.e., the slope of the function) of the linear-logistic function to zero. By solving this equality, it was determined that Rmax can be calculated with Equation (2) and Rmin can be calculated with Equation (3). The inflection points Rmax and Rmin will be used in the next step to describe two distinct physiological states in response to AA intake.
(2)$$\mathrm{Rmax}=-\frac{ln\left(\frac{-2b+cf+\sqrt{cf\left(cf-4b\right)}}{2bd}\right)}{f}$$
(3)$$\mathrm{Rmin}=-\frac{ln\left(\frac{-2b+cf-\sqrt{cf\left(cf-4b\right)}}{2bd}\right)}{f}$$

To investigate the relevance of the responses described by the linear logistic model to variables of growth, protein retention, and key physiological functions, a literature review was performed. Scientific studies whose data illustrated the linear-logistic model alongside other physiological response measures were selected. The scientific articles were chosen using the rapid review approach, which is similar to systematic reviews but with narrower search criteria [21]. The relevant data was extracted from the selected studies, but not conclusions, data interpretation, or data analysis performed by the authors. Articles published before August 2021 (search date) and available in the main databases for science research were considered (PubMed, ISI Web of Science, Science Direct, Scopus, and SciELO). The eligibility of the studies was determined under the following criteria:Only studies written in English were considered;Data from abstracts and oral presentations were not considered;Studies that provided AA in a wide-enough range to detect the linear-logistic pattern in protein retention and growth measurements were considered;Growth was considered an indirect measure of protein retention in stages other than gestation, as the products of conception are less protein-dense than lean tissue [22]; therefore, increases in body weight may not necessarily indicate increased protein retention during gestation;Only studies that measured the effect of AA intake on physiological responses related to reproduction, health, and survivability, in addition to growth or protein retention variables, were considered;Studies that measured plasma AA concentrations, in addition to growth or protein retention variables, were considered;Studies that reported methodological issues, such as unexplained responses caused by factors other than dietary AA intake, were not considered.Studies in which physiological responses did not follow a clear trend were considered to have random fluctuations (i.e., unexplained variance) and thus not considered.

CurveExpert Professional (version 2.7.3) was used to fit linear-logistic models for growth and protein retention measures in the selected studies. For physiological responses other than growth and protein retention, the model with the best goodness of fit as evaluated by the software was selected.

## 3. Results

The data mining component of the EDA resulted in the development of the linear-logistic model, which describes the response of growth or protein retention to varying levels of AA intake. The linear-logistic model describes the body weight gain response to dietary AA intake in various species, such as rats, fish, shrimp, and poultry (Figure 3). In addition, the pattern described by the linear-logistic model was observed in studies that measured protein retention using the IAAO technique (Figure 4).

In growing pigs, the weight gain response to AA intake followed the linear-logistic trend in dozens of studies, some of which are shown in Figure 5. The findings of these studies indicate that both the growth and protein retention responses to AA intake follow a linear-logistic trend and that this pattern is consistently observed in the published literature. Nonetheless, it is recommended to conduct systematic reviews to determine the extent to which the linear-logistic pattern occurs in different species and stages of growth.

The studies considered for investigating the physiological responses captured by the linear-logistic model are listed in Table 1 and detailed below. When data on the effects of rumen-protected methionine supplementation in Holstein cows [53] were fitted with a linear-logistic model, milk production was maximized at AA doses corresponding to the inflection point where weight gain was reduced, or Rmin (Figure 6). The increase in milk production occurred without a decrease in milk quality [53]; consequently, Holstein cows increased their daily production of milk fat and protein. Moreover, a reinterpretation of Ramirez-Camba et al. [54], which includes data on the effects of lysine intake on nitrogen retention and reproductive performance of 69 pregnant primiparous sows, revealed that dietary SID lysine levels associated with Rmin were related to increased litter size (Figure 7).

In the case of methionine supply in broiler breeder hens [56], Rmin coincides with an increase in immune response (Figure 8). In addition, in a study on the effects of valine intake on juvenile Nile tilapia [58], survivability was maximized at Rmin (Figure 9).

The reanalysis of Corzo et al. [26], who fed male broilers graded dietary arginine-to-lysine ratios, shows that dietary levels corresponding to Rmin resulted in the lowest mortality (Figure 10A). The data from Corzo et al. [26] show that the linear-logistic model was observed in animals fed from 1 to 14 days of age, but the pattern was no longer detectable once the animals reached 25 days of age (Figure 10B). It is possible that animals that received dietary arginine-to-lysine ratios corresponding to Rmin (117%) had an increased priority for health-related metabolic functions. These metabolic functions may have enhanced processes that resulted in a potential increase in AA absorption and/or AA bioavailability, thus matching the growth performance of the adjacent groups (107 and 127%), indicating a plateau; in fact, the Rmin group demonstrates improved feed-to-gain ratios in comparison to groups with a lower AA supply. Therefore, if the adaptation period to experimental diets is too long, the linear-logistic model’s predicted response may not be detectable. Additional research is necessary to determine the appropriate experimental durations for various essential AA, species, and production stages to detect the linear-logistic trend.

Data from Wellington et al. [60] on the effects of SID threonine on protein deposition and markers of gut health in colonic tissue in pigs fed different levels of dietary fiber and fermentable crude protein show greater gene expression of the zonula occludens protein-1 (ZO-1) at SID threonine doses associated with Rmin in comparison to Rmax (Figure 11A–D). Because the expression of ZO-1 is used as a marker of intestinal health, more specifically, intestinal permeability [61], it seems dietary threonine levels associated with Rmin promote gut barrier function in relation to threonine levels associated with Rmax. However, definitive conclusions cannot be drawn due to the small size of the data set (only three data points).

The data reported by Jayaraman et al. [57] on weaned pigs under unclean sanitary conditions and the data reported by Remus et al. [59] on growing pigs under a group-phase feeding system revealed increases in plasma Thr-to-Lys ratios at dietary Thr-to-Lys ratios approaching Rmin (Figure 12A,B, respectively). Similarly, the data reported by Lewis et al. [55] on pigs fed with 5% supplemental fat and the data reported by Reigh and Williams [62] on American alligators (*Alligator mississippiensis*) revealed increases in plasma Lys concentrations as dietary Lys levels approached Rmin (Figure 12C,D, respectively).

## 4. Discussion

The results of the EDA revealed significant differences in protein retention between animals of different genetic backgrounds. For example, studies from the 1990s found lower protein retention in pigs of the same age than studies from the 2010s. Nonetheless, the linear-logistic trend was found in papers from both decades. Genetics appears to influence maximum levels of protein retention but not the dynamics of AA utilization, which appear to be captured by the linear-logistic model. Similarly, the results of the EDA suggest that diets containing varying amounts of fiber and fermentable crude protein influence maximum protein retention but do not alter the dynamics of protein retention described by the linear-logistic model. As a result, no age, genetics, dietary fiber, or dietary fermentable crude protein correction appears to be required when fitting linear-logistic models.

Furthermore, the results of the EDA suggest that physiological functions associated with immune response, gut permeability, reproduction, and survival are further increased at AA levels above those for optimal growth. These results contradict the widely accepted concept of the hierarchy of nutrient use [17], according to which the body prioritizes survival and reproduction above growth and protein retention. Although these findings challenge current dogma in animal nutrition, they are consistent with findings from a variety of studies and response variables.

For example, Bhargava et al. [63] observed that when chicks challenged with the Newcastle virus were fed diets with increasing levels of valine, both growth and antibody titres improved, but the antibody titre response increased further after the animal had reached a plateau in its growth. The results of Bhargava et al. [63] suggest that minimum dietary valine levels for maximum growth are insufficient to optimize the immune response. In addition, Young and Marchini [20] and Robinson et al. [14] reported that, at dietary methionine intakes lower than those required for maximal protein synthesis, the rate at which methionine enters transmethylation reactions is reduced compared to the rate at which methionine enters protein synthesis pathways. That is, the body prioritizes protein synthesis in comparison to transmethylation reactions. As stated by Robinson et al. [14], a typical methionine requirement study that uses protein synthesis as the primary outcome is unlikely to detect the dietary needs for transmethylation reactions. Nevertheless, transmethylation reactions are important for the development and function of the gastrointestinal system [64,65], as well as the biosynthesis of metabolites such as taurine, which, besides being an essential AA in utero [66], regulates a variety of functions in the reproductive system, central nervous system, and renal system, among other important physiological functions [67,68,69,70,71]. Thus, traditional measures of protein synthesis seem to be insufficient to estimate methionine recommendations that optimize metabolic status. Similarly, when protein intake is lower than what is required for maximum protein synthesis, catabolism of branched chain AA is reduced, and protein synthesis is prioritized [20,72]. Nevertheless, catabolism of branched chain AA is important for the proper functioning and development of the central nervous system [73], placenta [74], mammary gland [75], and small intestine [76,77]. As a result, the body seems to prioritize the utilization of essential AA for protein synthesis, with surpluses being catabolized into a variety of metabolites that control key physiological processes [15,20]. It is noteworthy that AA levels above those required for protein synthesis requirements are catabolized or excreted [73], and that catabolism and excretion are not synonymous terms. Levels of AA that exceed the maximum required for protein synthesis are catabolized into functional metabolites, and levels beyond that are eliminated through excretion [20]. Under the linear-logistic approach, AA levels corresponding to Rmin may result in optimal biosynthesis of key metabolites and thus optimal metabolic status, a hypothesis that needs to be empirically tested.

Additionally, Marchini et al. [15] concluded that traditional measures of protein synthesis are equivocal and limited for establishing AA requirements when metabolic status is considered. The limitations described by Marchini et al. [15] are based on the body’s ability to maintain an AA and nitrogen balance by reducing important metabolic functions such as protein turnover [20]. This suggests that, as AA intake exceeds Rmax, turnover rates also increase, potentially resulting in improved physiologic and metabolic status. As stated by Young and Marchini [20], “a relatively high rate of muscle protein turnover presumably would favor the continued availability of glutamine for meeting lymphocyte function. Conversely, it can be speculated that a reduced rate of muscle protein turnover would diminish the availability of glutamine from this tissue source with adverse functional consequences”. Young and Marchini [20], based on observations made by Waterlow [78], also stated that reduced rates of protein turnover may compromise the body’s ability to resist trauma, infection, and thus stressful stimuli due to the potential reduction in glutamine flux. Consequently, it is probable that AA dosages corresponding to Rmin could maximize turnover rates and their associated metabolic benefits. The decrease in growth and protein retention observed at AA intake levels associated with Rmin may be explained by an increase in protein catabolism (turnover).

According to the EDA and the previously reviewed studies, animals prioritize lean tissue deposition over functions associated with long-term survival and reproduction, without discounting the AA required for short-term survival (i.e., maintenance), which must be met prior to lean deposition. In contrast, Hammond’s thesis asserts that low metabolically active tissues, such as lean and adipose tissues, had the least impact on co-ordinating body functions and animal survival, so the body assigned them fewer and fewer nutrients as nutrients became limited [79]. According to observations from research on wild boars, however, when nutrients become scarce, prioritizing the deposition of lean tissue can increase animal survival. According to Morelle et al. [80], wild boar restrict their movement in a rich nutritional environment, whereas in a poor nutritional environment, they increase their mobility in search of food. Therefore, when nutrient resources are scarce, wild boar survival rates may increase if the animals prioritize the preservation and deposition of lean tissue mass, which is required not only for the mobility necessary for foraging but also for evading a greater number of predators when expanding walking areas [81]. Prioritizing lean tissue deposition when nutrients are scarce may thus provide an evolutionary advantage to wild boar, and although this response is redundant in domesticated pigs, they may still exhibit it. Regardless of the reason, the responses observed in the current EDA correspond more closely to a conceptual model in which lean tissue mass is prioritized over metabolic status and reproduction.

This exploratory study challenges not only traditional measures of growth and protein retention for AA requirement estimation (and their associated statistical models), but also traditional plasma AA concentration measurements. The main assumption in the use of the plasma AA technique for estimating requirements is that AA doses that result in increased concentrations of plasma AA are excessive [19]. The results of the present study indicate, however, that plasma AA concentrations are maximized at dietary AA levels corresponding to Rmin, levels that are in turn associated with improvements in key physiological responses and reproduction. In this regard, multiple authors have speculated that increased free plasma AA concentrations result in increased AA availability for the immune response and other physiological functions [82,83,84,85,86,87]. Additionally, research on pregnant sows has shown that dietary arginine supplementation increased plasma AA concentration, which was associated with an increase in the number of live-born piglets and litter birth weight [88]. Other authors have also reported improvements in fetal development with associated increments in plasma AA concentrations in sows [89,90,91]. Similarly, Wang et al. [92] reported that leucine-supplemented doses that resulted in a peak in plasma leucine concentration during late gestation in sows also resulted in a reduction in stillborn and mummified piglets, and increased piglet birth weight. Thus, the findings of the EDA and the previously mentioned studies suggest that increased plasma AA concentrations are associated with improved metabolic status and reproduction rather than being excessive, as previously believed.

This exploratory study proposes a new paradigm for the estimation of AA requirements that, despite requiring further evaluation, has the potential to increase the efficiency of animal production. Under the linear-logistic approach, dietary AA levels that correspond to Rmin have been shown to improve key physiological responses, reproduction, and survival. Thus, for the stages of production where the main goal is to improve reproductive performance and animal survival and robustness, AA doses corresponding to Rmin may be considered as the requirement. For production stages where the primary goal is to maximize growth rates (e.g., finishing pigs), AA doses corresponding to Rmax may be considered the recommended dose. The use of the linear-logistic model in AA requirements research has the potential to result in nutritional strategies that increase reproductive efficiency, enhance animal health and welfare, and ultimately increase productivity across different livestock species by reducing mortality and removal rates.

## 5. Conclusions

In conclusion, the current EDA results suggest that AA requirements estimated as minimum dietary AA levels for maximum growth and protein retention are insufficient to develop dietary recommendations that optimize key physiological responses associated with survival and reproduction. In addition, the EDA results suggest that growth and protein retention measurements can be used to indirectly estimate AA doses that optimize key physiological responses such as reproduction and, conceivably, survival rates when growth and protein retention responses are analyzed using the linear-logistic model. However, this assertion must be confirmed by additional research.

## Figures and Tables

**Figure 1 animals-13-01708-f001:**
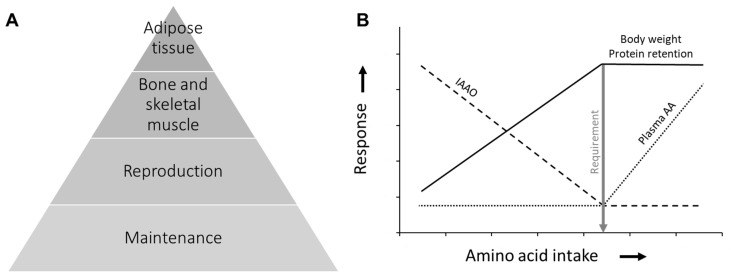
Current paradigm in amino acid (AA) requirement estimation: (**A**) pyramid chart of the hierarchical model of nutrient use proposed by Hammond [17]; (**B**) response patterns to increasing intake of a limiting AA in accordance with current methods for estimating AA requirements, adapted from [18]. IAAO: indicator AA oxidation.

**Figure 2 animals-13-01708-f002:**
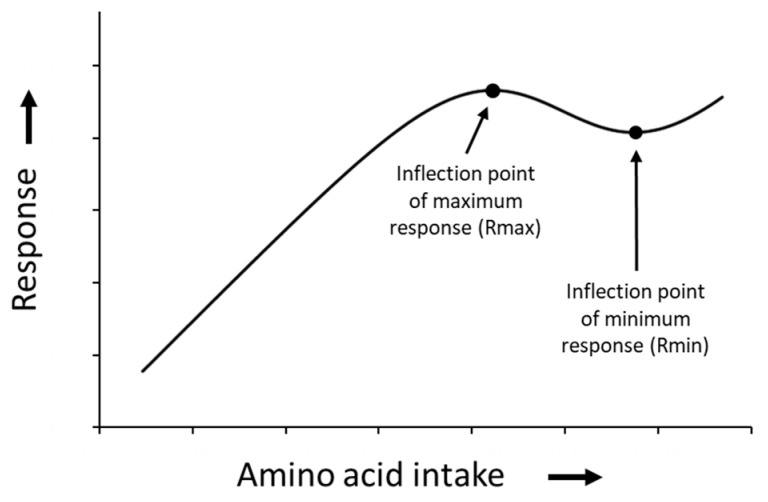
The linear-logistic model depicts two inflection points: the point at which the response is maximized (Rmax) and the point at which the response is minimized (Rmin).

**Figure 3 animals-13-01708-f003:**
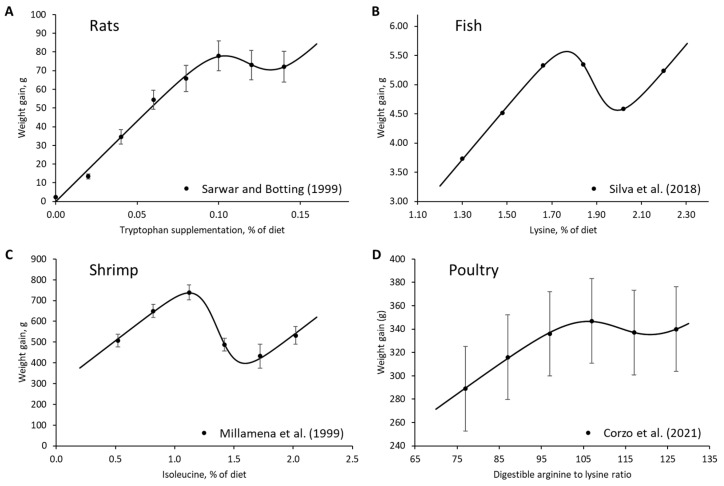
The linear-logistic model was observed to describe the weight gain response to amino acid intake in various species, such as (**A**) rats—data extracted from Sarwar and Botting (1999) [23]; (**B**) fish (*Colossoma macropomum*)—data extracted from Silva et al. (2018) [24]; (**C**) shrimp (*Penaeus monodon*)—data extracted from Millamena et al. (1999) [25]; and (**D**) poultry (male broilers)—data extracted from Corzo et al. (2021) [26].

**Figure 4 animals-13-01708-f004:**
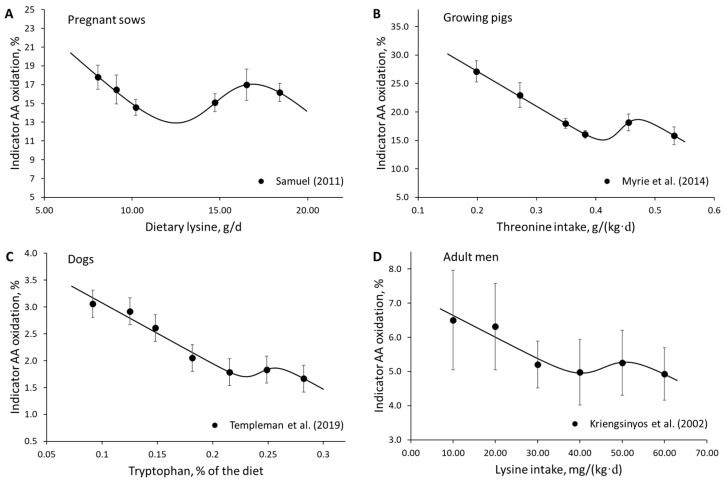
The linear-logistic model was observed to describe the protein synthesis response, measured using the indicator amino acid oxidation technique in (**A**) pregnant sows—data extracted from Samuel (2011) [27]; (**B**) growing pigs—data extracted from Myrie et al. (2014) [28]; (**C**) dogs (Labrador Retrievers)—data extracted from Templeman et al. (2019) [29]; and (**D**) healthy adult men—data extracted from Kriengsinyos et al. (2002) [30].

**Figure 5 animals-13-01708-f005:**
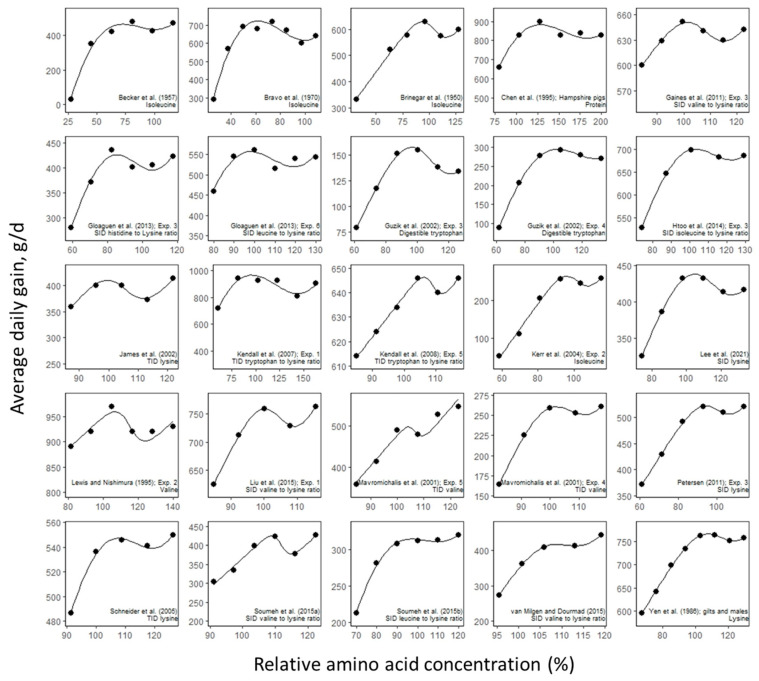
The trend described by the linear-logistic model can be observed in different studies on the effect of protein and amino acid (AA) intake on average daily gain in growing pigs. The protein and dietary AA concentrations are expressed as a relative AA concentration (percentage) of the requirements estimated by NRC (2012), calculated as [AA concentration (%)/AA requirement (%)] × 100 wi. Each plot indicates the study from which the data were extracted and the supplemented nutrient. Data extracted from references [31,32,33,34,35,36,37,38,39,40,41,42,43,44,45,46,47,48,49,50,51,52]. Exp.: experiment; SID: standardized ileal digestible; TID: true ileal digestible.

**Figure 6 animals-13-01708-f006:**
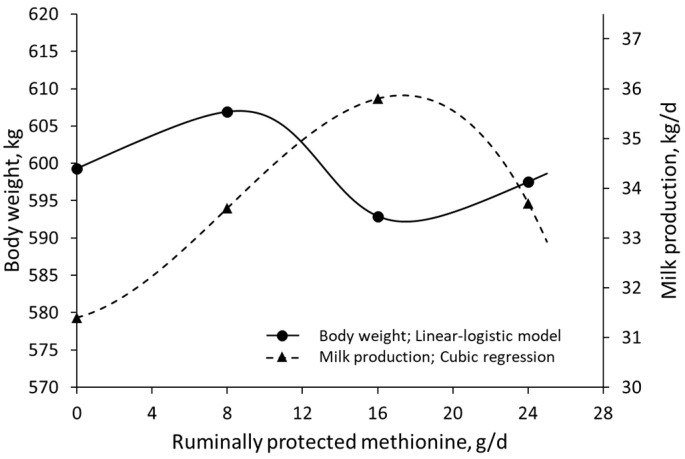
Effects of rumen-protected methionine supplementation on body weight and milk production in Holstein cows fitted with a linear-logistic model and segmented regression, respectively; data extracted from Lara et al. (2006) [53].

**Figure 7 animals-13-01708-f007:**
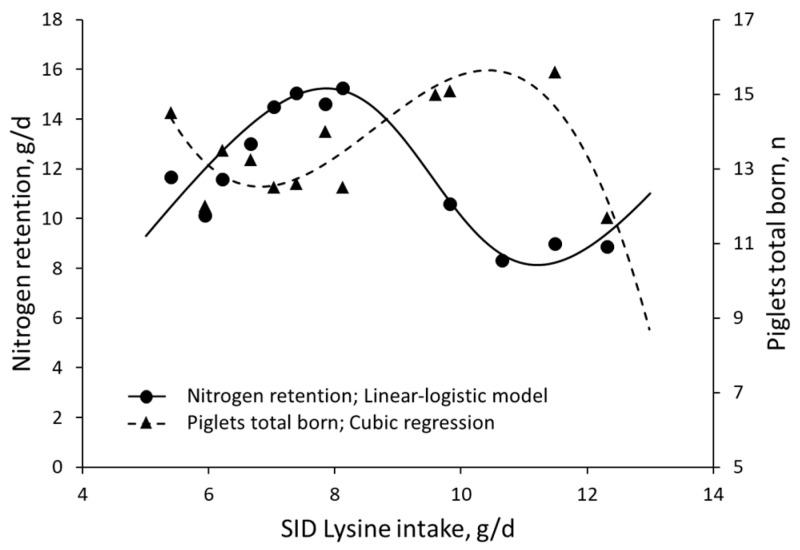
A reinterpretation of Ramirez-Camba et al. (2020) [54] shows a reduction in nitrogen retention at an SID lysine intake associated with an increase in the total number of pigs born in pregnant sows fed experimental diets at d 50 of gestation.

**Figure 8 animals-13-01708-f008:**
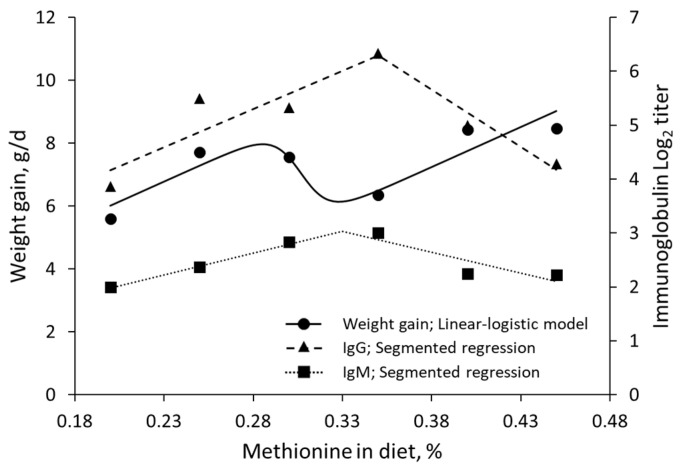
Performance and immune responses to graded levels of methionine intake in broiler breeder hens fitted with a linear-logistic model and segmented regression; data extracted from Hosseini et al. (2011) [56].

**Figure 9 animals-13-01708-f009:**
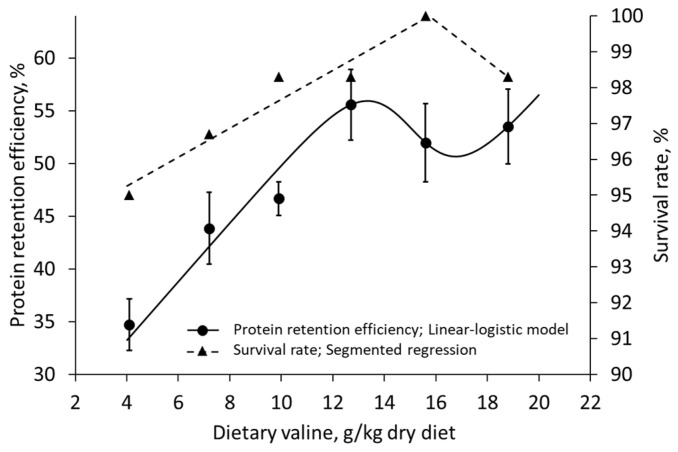
Effects of valine intake on protein retention efficiency and survival rates on juvenile Nile tilapia (*Oreochromis niloticus*) fitted with the linear-logistic model and segmented regression, respectively; data extracted from Xiao et al. (2018) [58].

**Figure 10 animals-13-01708-f010:**
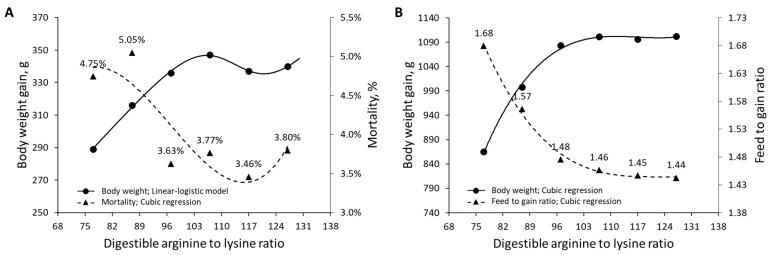
(**A**) Effect of dietary digestible arginine-to-lysine ratios on body weight gain in Ross 708 male broilers from 1 to 14 days of age and mortality rates from 1 to 25 days of age. (**B**) Effect of dietary digestible arginine-to-lysine ratios on body weight gain and feed to gain ratios in Ross 708 male broilers from 1 to 25 days of age; data extracted from Corzo et al. (2021) [26].

**Figure 11 animals-13-01708-f011:**
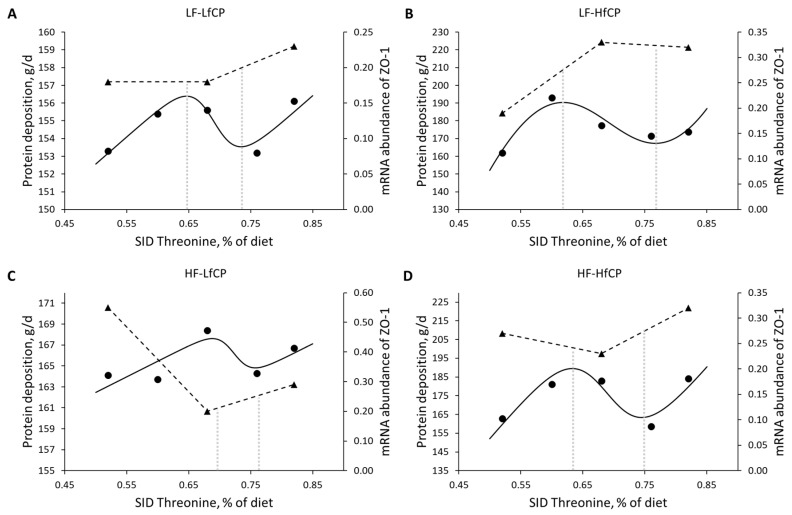
Effects of SID threonine on protein deposition and the expression of the zonula occludens protein-1 (ZO-1) gene in colonic tissue in pigs fed: (**A**) Low fiber and low fermentable crude protein. (**B**) Low fiber and high fermentable crude protein. (**C**) High fiber and low fermentable crude protein. (**D**) High fiber and high fermentable crude protein. Data extracted from Wellington et al. (2020) [60]. The protein deposition response (●) was fitted with linear-logistic models (solid lines), and the mRNA abundance of the ZO-1 gene (▲) was fitted with segmented regression (dashed lines). HF: high fiber; HfCP = high fermentable crude protein; LF: low fiber; LfCP = low fermentable crude protein.

**Figure 12 animals-13-01708-f012:**
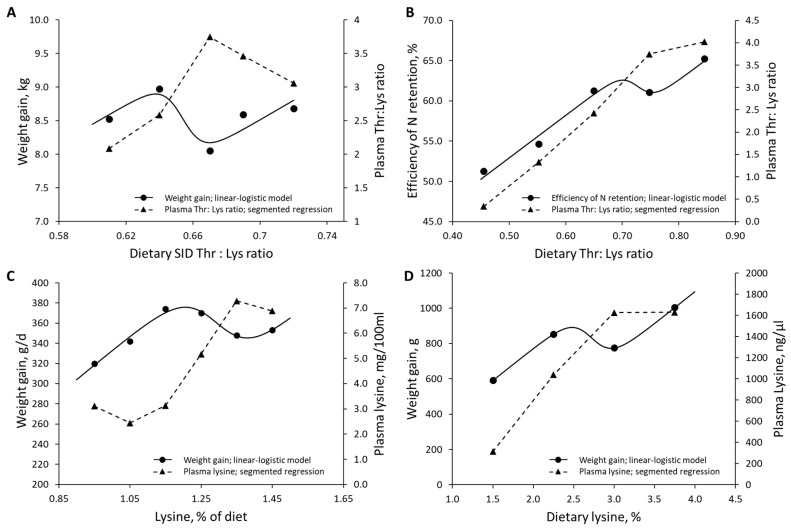
(**A**) Effects of dietary SID threonine (Thr)-to-lysine (Lys) ratios on weight gain and plasma Thr-to-Lys ratios on weaned pigs under unclean sanitary conditions; data extracted from Jayaraman et al. (2015) [57]. (**B**) Effects of dietary Thr-to-Lys ratios on the efficiency of nitrogen (N) retention and plasma Thr-to-Lys ratios on growing pigs under a group-phase feeding system; data extracted from Remus et al. (2019) [59]. (**C**) Effect of dietary Lys intake on weight gain and plasma Lys concentration in pigs with 5% supplemented fat; data extracted from Lewis et al. (1980) [55]. (**D**) Effects of dietary Lys intake on weight gain and plasma Lys concentration in the American alligator (*Alligator mississippiensis*); data extracted from Reigh and Williams (2022) [62].

**Table 1 animals-13-01708-t001:** Studies in which growth or protein retention measurements followed a linear-logistic trend and reported additional physiological responses.

Publication	Animal	Amino Acid	Dose Range	Protein Accretion Variable	Physiological Response Variable
Lewis et al. (1980) [55]	Growing pigs	Lys	0.95–1.45% of the diet	Weight gain	Plasma AA
Lara et al. (2006) [53]	Holstein cows	RP Met	0–24 g/d	Body weight	Milk production
Hosseini et al. (2011) [56]	Broiler hens	Met	0.2%–0.45% of the diet	Weight gain	IgG & IgM
Jayaraman et al. (2015) [57]	Growing pigs	Lys	61–72 SID Thr: Lys ratio	Weight gain	Plasma AA
Xiao et al. (2018) [58]	Nile tilapia	Val	4.1–18.8 g/kg dried diet	Protein retention	Survival rate
Remus et al. (2019) [59]	Growing pigs	Thr	46–85 Thr to Lys ratio	N retention	Plasma AA
Ramirez-Camba et al. (2020) [54]	Pregnant sows	SID Lys	5.4–12.3 g/d	N retention	Total piglets born
Wellington et al. (2020) [60]	Growing pigs	SID Thr	0.52–0.82% of the diet	Protein retention	ZO-1 ^1^
Corzo et al. (2021) [26]	Male broilers	Arg	77–127 dArg: Lys ratio	Body weight	Survival rate

AA: amino acid; Arg: arginine; dArg: digestible arginine; IgG: immunoglobulin G; IgM: immunoglobulin M; Lys; lysine; Met: methionine; N: nitrogen; RP: rumen-protected; SID: standardized ileal digestible; Thr: threonine; Val: valine; ZO-1: zonula occludens protein-1. ^1^ mRNA abundance of the ZO-1 gene in colonic tissue.

## Data Availability

The data used is available in each of the papers cited and displayed in the plots. No new data was created.

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
