# Peer review of "The Linear-Logistic Model: A Novel Paradigm for Estimating Dietary Amino Acid Requirements"

_animals, 2023, doi:10.3390/ani13101708_

Round 1

Reviewer 1 Report

Manuscript ID - animals-2339716

The linear-logistic model: a novel paradigm for estimating dietary amino acid requirements.

General comments

 ·         Did you consider the difference in genetics, especially in pigs? This may have a significant impact on the lean growth potential, which will affect the AA requirement response.

·         How did you account for the response affected by age? Growing pig’s response to AA might be different than piglets post-weaning.

·         In Figure 5, you have represented the x-axis as the relative intake of the various AA. Are these values really the relative intakes or the dietary concentrations of the test AA?

·         Figure 5: To bring clarity, relative intake will be a measure of actual feed intake vs. the dietary AA concentration. If the animals are restrictively fed to determine requirements, as is done in most cases, this will be difficult to report as relative intake because we cannot estimate accurately what the voluntary feed intakes will be should the animals be allowed ad-lib feeding.

·         In your conclusion, you suggest that Rmin levels on AA may be what is required to optimize physiological response (L376-379). However, since Rmin is a lower AA requirement level than the Rmax, why do we need to have 2 separate AA requirements when Rmax levels for an AA is achieved, this already includes the lower Rmin value. How do we explain this practically to a producer who needs both a healthy and production animal? Why not suggest Rmax solely?

In Table 5, it is important to stress that the response of animals to dietary AA levels can be affected by the diet intrinsic factors, such as the level of fiber in the diet and the level of ANF in protein sources such as soybean meal. Did the model consider studies where high-fiber diets were tested in response to a test AA and what effects it had on growth or protein retention?

Check these papers:

Mathai et al., 2016: JAS 94(10): 4217-4230

https://doi.org/10.2527/jas.2016-0680

Wellington et al., 2018: JAS 96(12): 5222-5232

https://doi.org/10.1093/jas/sky381

Specific comments

L101, repetition, please delete.

L131, you need to state the total number of studies that were finally considered after the eligibility screening.

L274-275, It is correct these physiological factors increase AA requirements. However, your assertions on L276-277 cannot be right.

This is because the traditional approach used in these studies always depends on response parameters, such as growth or protein deposition. This assumes that when priory functions are considered first (maintenance) the surplus AA are used for protein deposition. Therefore, studies have not focused on directly determining AA requirements for physiological functions but rather indirectly. This can be confirmed in publications looking at the impact of immune challenge on AA requirements, an example is Jayaraman et al., 2017 which you cited in the manuscript.

L284-285 This is in-line with the push and pull effect. If you supply more AA than required for a response parameter like growth, the body will utilize this extra AA for immune enhancement, not necessarily because it requires that for survival, but as a conservational means. Because it costs more energy to catabolism AA for excretion than incorporating in cytokine build-up, for example, because these functions are happening de novo, anyway. This note here concurs with L309-310.

L336-340 lean tissue deposition seems prioritized because the design of the studies have largely focused on lean deposition as output parameters. However, we cannot discount the maintenance requirement threshold that needs to be met before lean deposition. Any extra AA will be further utilized, but that doesn’t necessarily mean they are absolutely required for survival. Please rephrase this section of your manuscript.

Reviewer 2 Report

This study determined whether current methods for estimating AA requirements for animal health and welfare are sufficient. An exploratory data analysis (EDA) was conducted, which involved a review of assumptions underlying AA requirements research, a data mining approach to identify animal responses to dietary AA levels exceeding those for maximum protein retention, and a literature review to assess the physiological relevance of the linear-logistic model developed through the data mining approach. The results suggest that current methods based solely on growth and protein retention measurements are insufficient for optimizing key physiological responses associated with health, survival, and reproduction. The linear-logistic model could be used to estimate AA doses that optimize these responses, and potentially survival rates. The topic is worth studying and the manuscript is well written. However, there are several grammatical errors and formatting errors in the text. Therefore, it is recommended that the English expression of the article be revised carefully. In addition, a number of issues need to be addressed to further improve the manuscript.

L162: Change “indicator AA oxidation (IAAO) technique” to “IAAO technique”. Abbreviations do not require repeated explanations

L267-269: Thr to Lys ratios concentration?

L281-286: Delete this paragraph because it is repeated with the following text.

L368: The reference [92] is not included in the literature list.

According to the requirements of the submission journal, revise the reference format uniformly.

Round 2

Reviewer 1 Report

I am satisfied with the response of the authors.

I recommend publication.

Author Response

English language, style, and spell checks were performed.